# Utilising multi-modal data-driven network analysis to identify monotherapy and combinational therapy targets in SOX2-dependent squamous cell lung cancer

Woochang Hwang[1,2,10,11], Daniel Kottmann [3,11], Wenrui Guo[3], Méabh MacMahon[1,2,4], Lucia Correia[3,10], Sherine Ahmed[3], Rebecca Harris[1,10], Frank McCaughan [3,12] ✉ & Namshik Han [1,5,6,7,8,9,12] ✉

Drug discovery requires understanding disease mechanisms, making the integration of multi-modal data essential. These data types, including omics, disease-associated, and pathway information, must be combined to uncover therapeutic insights. We developed iPANDDA, a computational pipeline that integrates these data through a network-based approach to predict candidate drug targets for specific diseases. We applied iPANDDA to lung squamous cell carcinoma (LUSC), a subtype of non-small cell lung cancer representing ~25% of global cases. Despite advances in cancer therapeutics, targeted treatments for LUSC remain limited, partly due to a lack of robust models to study carcinogenesis and therapeutic response. The SOX2 gene, amplified in ~50% of patients, plays a critical role in sustaining the cancer phenotype. Using iPANDDA, we identified and validated SOX2-dependent therapeutic targets. In vitro inhibition studies confirmed AKT and mTOR complexes as key monotherapy and combination therapy targets and revealed pathways for SOX2-targeted combination therapies.

Drug discovery is a lengthy, costly process with a high failure rate, often due to insufficient efficacy in the target disease[1]. This failure frequently stems from a limited understanding of disease pathogenesis and the role of specific drug targets in the disease. The increasing availability of datasets on individual diseases and potential drug targets has spurred efforts to integrate disparate biological data to enhance target identification and streamline drug development[1,2]. Additionally, combination therapies are gaining attention for their potential to improve therapeutic outcomes[3–5]. However, integrating diverse datasets to predict effective monotherapies and combination therapies remains a significant challenge, highlighting the unmet need for computational algorithms capable of prioritizing therapeutic targets in a disease-specific context.

Lung cancer is the leading cause of cancer-related deaths worldwide. Lung squamous cell carcinoma (LUSC), a subtype of non-small cell lung cancer (NSCLC), accounts for approximately 25% of all lung cancer cases[6]. Despite advances in cancer therapeutics, no targeted therapies have been approved for LUSC. Among its key molecular drivers, amplification of the 3q26 locus is observed in about 50% of LUSC patients[7–10]. Evidence suggests that SOX2 is a critical target gene of this amplification and a key molecular driver of LUSC pathogenesis. However, targeting transcription factor s (TFs) like SOX2 poses significant therapeutic challenges due to their promiscuous interactions with other proteins and DNA binding sites. These complexities have hindered efforts to define the critical upstream and downstream proteins or

[1]Milner Therapeutics Institute, University of Cambridge, Cambridge, UK. [2]CardiaTec Biosciences LTD, Cambridge, UK. [3]Victor Phillip Dahdaleh Heart and Lung Research Institute, Department of Medicine, University of Cambridge, Cambridge, UK. [4]Centre for Therapeutics Discovery, LifeArc, Stevenage, UK. [5]Cambridge Centre for AI in Medicine, Department of Applied Mathematics and Theoretical Physics, University of Cambridge, Cambridge, UK. [6]Cambridge Stem Cell Institute, University of Cambridge, Cambridge, UK. [7]Department of Quantum Information, Institute for Convergence Research and Education in Advanced Technology and Engineering, Yonsei University, Seoul, Republic of Korea. [8]Department of Nano Biomedical Engineering (NanoBME), Advanced Science Institute, Yonsei University, Seoul, Republic of Korea. [9]Center for Nanomedicine, Institute for Basic Science (IBS), Seoul, Republic of Korea. [10]Present address: WH and MM - CardiaTec Biosciences, Cambridge, UK; LC - Adaptimmune, Abingdon, UK; RH - Cancer Research Horizons, Cambridge, UK. [11]These authors contributed equally: Woochang Hwang, Daniel Kottmann. [12]These authors jointly supervised this work. Frank McCaughan. Namshik Han. ✉e-mail: fm319@cam.ac.uk; nh417@cam.ac.uk

pathways influenced by SOX2, resulting in the absence of targeted treatments for this devastating disease.

Recent advances in molecular profiling have generated a wealth of publicly available 'omics' datasets for LUSC[9], along with functional datasets from CRISPR and RNAi screens[11]. However, integrating these complex and heterogeneous datasets to derive actionable insights remains a formidable task. Sophisticated analytical tools that integrate multi-modal datasets are urgently needed to comprehensively understand the pathobiology of LUSC and identify innovative therapeutic strategies.

To address these challenges, we developed iPANDDA (in-silico Pipeline for Agnostic Network-based Drug Discovery Analysis), a computational algorithm designed to integrate and analyse multi-modal datasets. Using iPANDDA, we constructed a meaningful protein-protein interaction (PPI) network centred on SOX2 and LUSC, identifying therapeutically vulnerable nodes. These targets were then ranked based on disease relevance and druggability.

This systematic approach identified candidate targets for SOX2-dependent LUSC, which we validated experimentally. iPANDDA successfully identified known therapeutic targets, thus validating our approach, as well as identifying a number of promising previously unreported targets and drug candidates for this unmet clinical need.

## Results

### iPANDDA: a pipeline to identify candidate drugs and druggable pathways for diseases associated with specific genes of interest

iPANDDA is a pipeline developed to identify candidate drugs and druggable pathways for diseases with a specific genetic dependency but for which treatment options are limited. We developed iPANDDA using an agnostic data-driven approach to construct a network-based model of the disease of interest with the aim of identifying key disease nodes and potential drug targets. The input of the iPANDDA pipeline is multi-modal data related to the disease; the outputs are candidate drugs and targets or networks that should be a focus of therapeutic efforts.

In this report, the focus is on a subset of LUSC that is SOX2-dependent and for which there are no specific therapeutics available. The pipeline is summarised in Fig. 1. There are five key computational steps prior to wet lab validation experiments. These are summarised below, with further details provided in the Methods section. The novelty of iPANDDA lies in the breadth of data feeding into the algorithm, its integration, and the incorporation of experimental data specifically to disease being examined, in this case SOX2-dependent LUSC.

1. **Multi-modal datasets related to SOX2-dependent LUSC:** iPANDDA utilises proprietary and publicly available multi-modal data associated with LUSC, comparative data from normal tissue, and additional proprietary experimental data.
   - Proprietary data are derived from an in vitro organotypic model system in which the driver oncogene SOX2 is acutely deregulated in

an immortalised airway epithelial cell line. This model aims to recapitulate the molecular, cellular, and physiological environment in which LUSC develops.
   - Public datasets include cancer-specific genomic and transcriptomic data from The Cancer Genome Atlas (TCGA)[12], normal tissue genomic and transcriptomic data from the Genotype-Tissue Expression (GTEx) database[13], and data from the Open Targets and OmniPath databases[14]. Open Targets focuses on disease-target associations, while OmniPath is focused on cell signalling networks. Additionally, data from the ENCODE database[15] on transcription factor (TF) activity and binding sites for SOX2 are incorporated.

2. **Data integration and network construction:** This step integrates all the information gathered from the multi-modal disease association study (Step 1) into a single data space, forming a network (graph-based data structure). The network includes PPIs of genes of interest, disease-associated genes from Open Targets, known drug targets, TFs for genes of interest, and differentially expressed genes (DEGs) from integrated transcriptomics data. This integrated network serves as the foundation for constructing a disease-specific network that encapsulates the entire disease signature.

3. **Network analysis:** The network is analysed to identify key proteins that are central to the SOX2 network and, therefore, may represent ideal therapeutic targets. Key proteins are determined using multiple network centrality algorithms and the Random Walk with Restart (RWR) algorithm. CRISPRi data from the Dependency Map (DepMap)[16] is used to further refine the network to retain only those with demonstrable LUSC dependency.

4. **In-silico drug simulation:** This step stratifies potential candidate drugs using a network proximity method[17] to predict how closely drug targets correspond to the identified key proteins in the network.
   - Druggable targets are those for which an existing compound to target[18] them exists based on the proximity analysis.
   - Undruggable targets are those for which no existing compound is available, as suggested by the drug proximity analysis.

5. **Target prioritisation:** Using iPANDDA, we rank druggable and undruggable targets according to a stepwise network analysis score. Prioritisation is based on a combination of factors, including the network analysis score, druggability assessment, structural clustering, RNA expression levels, and the results from the in-silico drug simulation. One of the advantages of creating a network is that it may suggest potential combinations of therapeutics targeting separate nodes within the network. Combination therapeutics are prioritised based on their potential for targeting more than one functional pathway within the network.

6. **In vitro validation:** We evaluate the impact of therapeutic compounds on SOX2-dependent and non-SOX2-dependent cell lines using a range

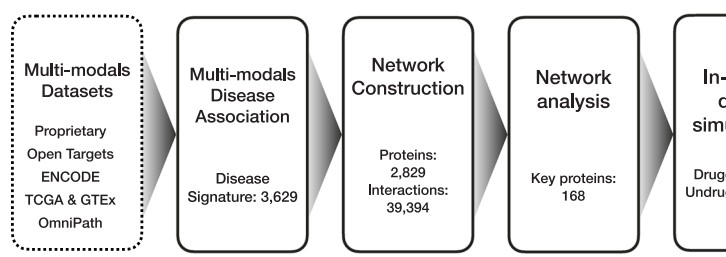
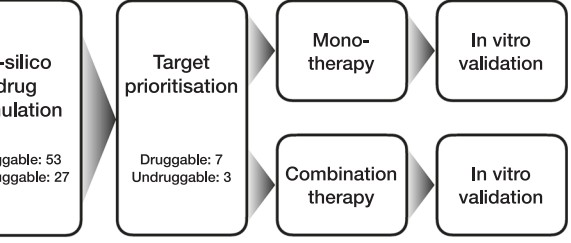

**Fig. 1 | Overview of iPANDDA.** iPANDDA is a computational pipeline designed to identify candidate drugs and therapeutic pathways for diseases of interest. For SOX2-dependent LUSC, multi-modal datasets were integrated to construct a disease-specific protein-protein interaction (PPI) network. Data sources included transcriptional signatures (1636 genes identified from LUSC-specific TCGA analysis, 2312 DEGs from a bespoke SOX2-dependent model using RNA-seq and ChIP-seq), transcription factors (12 from ENCODE), LUSC dependency data (157 from Open Targets), and SOX2-interacting proteins (12 from OmniPath). Network analysis using centrality algorithms and Random-Walk identified 168 core proteins. Drug simulation was applied to these core proteins to classify them as druggable or undruggable. Candidate drug-targets were prioritised using a combination of network and omics analyses, leading to in vitro validation experiments for 7 monotherapy candidates and the identification of previously unexplored combination therapy strategies.

of concentrations, analysing cell viability with a resazurin-based assay, and presenting results as dose-response curves.

### Identification of a multi-modal disease signature for constructing SOX2-dependent LUSC network

We interrogated multiple datasets associated with SOX2-dependent LUSC to establish a multidimensional disease-specific signature.

**LUSC/SOX2 transcriptional and functional signature.** We utilized proprietary RNA-seq datasets and the TCGA database to identify transcriptional signatures specific to SOX2-dependent LUSC. In the proprietary RNA-seq dataset, SOX2 was confirmed as the most highly upregulated gene, and DEGs were identified (Fig. 2A). To further analyse the regulatory role of SOX2 in LUSC, we incorporated SOX2 ChIP-seq datasets, which revealed direct binding sites and provided insights into its transcriptional regulatory network (Fig. 2B).

From the TCGA database, we conducted a cross-comparison of LUSC with the remaining 30 cancer types in the pan-cancer dataset to identify distinctly differentially expressed genes in LUSC. To enhance robustness, we integrated normal samples from TCGA and GTEx databases, overcoming the limitations of insufficient normal samples in the TCGA dataset (Supplementary Fig. 1). Instead of raw expression values, we used fold-change values to better capture differential expression. SOX2 showed significantly upregulated expression in LUSC compared to normal tissue (Fig. 2C). Using an ANOVA test ($p$-value ≤ 1e-150; see Methods), we identified 1636 genes with enriched differential expression in LUSC (Supplementary Table 1). Hierarchical clustering of the pan-cancer TCGA dataset using these 1636 genes showed that LUSC could be readily distinguished from other tumour types, with the nearest clusters comprising foregut-origin cancers, including lung adenocarcinoma (LUAD), head and neck squamous cell carcinoma (HNSC), and esophageal carcinoma (ESCA) (Fig. 2D). Further validation using the Human Protein Atlas enrichment test demonstrated that these genes are significantly associated with lung-specific biology (Supplementary Table 2).

Using the Open Targets database, we confirmed that SOX2 is predominantly associated with neoplasms (Fig. 2E). Additionally, we incorporated 62 LUSC-associated genes with an association score ≥0.8. This threshold was optimized using disease enrichment tests across gene lists, yielding the lowest $p$-value (6.0E-34) for LUSC. Furthermore, target proteins for drugs in Phase 3 or higher clinical trials for LUSC were also retrieved from the Open Targets database for inclusion in the analysis.

**SOX2 dependency signature.** Next, we focused on SOX2 dependency by analysing DEGs from a proprietary organotypic model (OTC) and publicly available datasets, as well as interrogating existing datasets for the SOX2 interactome or predicted dependency. In the OTC model, SOX2 deregulation in immortalized human bronchial epithelial cells identified 2,312 DEGs ($|logFC| \geq 1$, FDR < 0.05). These genes were significantly enriched in epithelial carcinomas, particularly squamous subtypes, as shown by the DisGeNet[19] over-representation protocol (Fig. 2F).

We then explored proteins directly interacting with SOX2 or serving as putative upstream transcriptional regulators of SOX2. From the Omnipath database, we identified 12 proteins with direct protein-protein interactions (PPIs) involving SOX2 (Fig. 2G). To identify upstream transcriptional regulators of SOX2, we used the ENCODE Transcription Factor Binding Sites (TFBS) database. A transcription factor (TF) over-representation analysis ($p$-value < 0.01) identified 12 TFs, including YY1, MYC, and CREB1. Fig. 2H illustrates an example of TFBS mapped to the PTEN gene, highlighting how these regulators influence key downstream targets. The regulatory map of the identified 12 TFs and their associated target genes is shown in Fig. 2I, emphasizing the extensive transcriptional control exerted by these TFs in SOX2-dependent LUSC.

### Construction of LUSC SOX2-dependent (LUSOX) network

To construct a comprehensive SOX2-dependency network for LUSC, we integrated disparate datasets encompassing transcriptional, functional, and interaction profiles (Supplementary Table 1). This network serves as a foundation to unravel SOX2's multifaceted role in LUSC pathogenesis and to identify potential therapeutic targets. We constructed a protein–protein interaction (PPI) network using the STRING database[20], resulting in a network comprising 2829 proteins and 39,394 interactions. Within this network, SOX2 directly interacts with 156 proteins, which formed the basis for all subsequent analyses, including community detection, network centrality analysis, and target prioritisation. Among these, 12 high-confidence interactions were also present in the OmniPath database, reflecting the overlap between STRING and more stringently curated PPI sources. These directly interacting proteins connect further with their first-order neighbours, forming a broader network of interactions. These neighbour proteins also interact with LUSC-associated genes, culminating in a network that includes 1234 genes, approximately half of the entire network. This extensive connectivity demonstrates SOX2's significant influence on the network through direct interactions, neighbour proteins, and connections to LUSC-associated genes. These findings underscore SOX2's central role in LUSC pathogenesis and its potential as a key target for therapeutic intervention.

### Unveiling of LUSC disease mechanisms by network community detection

To elucidate the influence of SOX2 on LUSC disease mechanisms, we used a network community detection method (Louvain protocol – see Methods) to group the LUSOX network into eight distinct communities, each related to specific pathways (Supplementary Fig. 2A). These communities correspond to key biological processes such as RHO GTPase signalling, the Wnt pathway, keratinization, cytokine signalling, transcription pathways, interferon signalling, the cell cycle, and lipid metabolism, all of which are implicated in LUSC pathogenesis[21,22].

We visualised these interdependencies using a Circos plot (Supplementary Fig. 2B–F), which highlights the intricate network of biological pathways influenced by SOX2. The plot illustrates interactions between SOX2 and its directly interacting proteins (PPI), LUSC master regulators (Master), target proteins of LUSC drugs (Drug), transcription factors (TF) of LUSC signature genes, and differentially expressed genes in key pathways. These key pathways were labelled C1 through C8, with each representing a unique functional role: RHO GTPase signalling (C1), the WNT pathway (C2), keratinization (C3), cytokine signalling in the immune system (C4), transcription pathways (C5), interferon signalling (C6), the cell cycle (C7), and lipid metabolism (C8). SOX2 not only directly interacts with key proteins within these pathways but also exerts indirect control through the "Masters," "Drug Targets," and TFs. The arcs connecting the clusters (C1 to C8) emphasize SOX2's pleiotropic role across multiple pathways implicated in LUSC, consistent with preclinical data suggesting it acts as a master "switch" for LUSC[21–23].

These pathways include critical processes such as the Wnt pathway, which has a complex, context-dependent regulatory relationship with SOX2 and is strongly implicated in development and carcinogenesis. Similarly, RHO GTPases are key signalling molecules associated with cancer hallmarks, while deregulation of the cell cycle is a defining feature of cancer. Keratinization is a hallmark histopathology feature of LUSC. The network also implicated key immunological pathways, including cytokine signalling and interferon signalling, which could shape the tumour microenvironment (Supplementary Fig. 2). This comprehensive network analysis confirms that SOX2's influence spans multiple interconnected routes, potentially opening new avenues for therapeutic intervention.

### Discovery of drug targets for SOX2-dependent LUSC by network analysis

To further refine the best targets in SOX2-dependent LUSC, we applied multiple network analysis algorithms to the LUSOX network, including

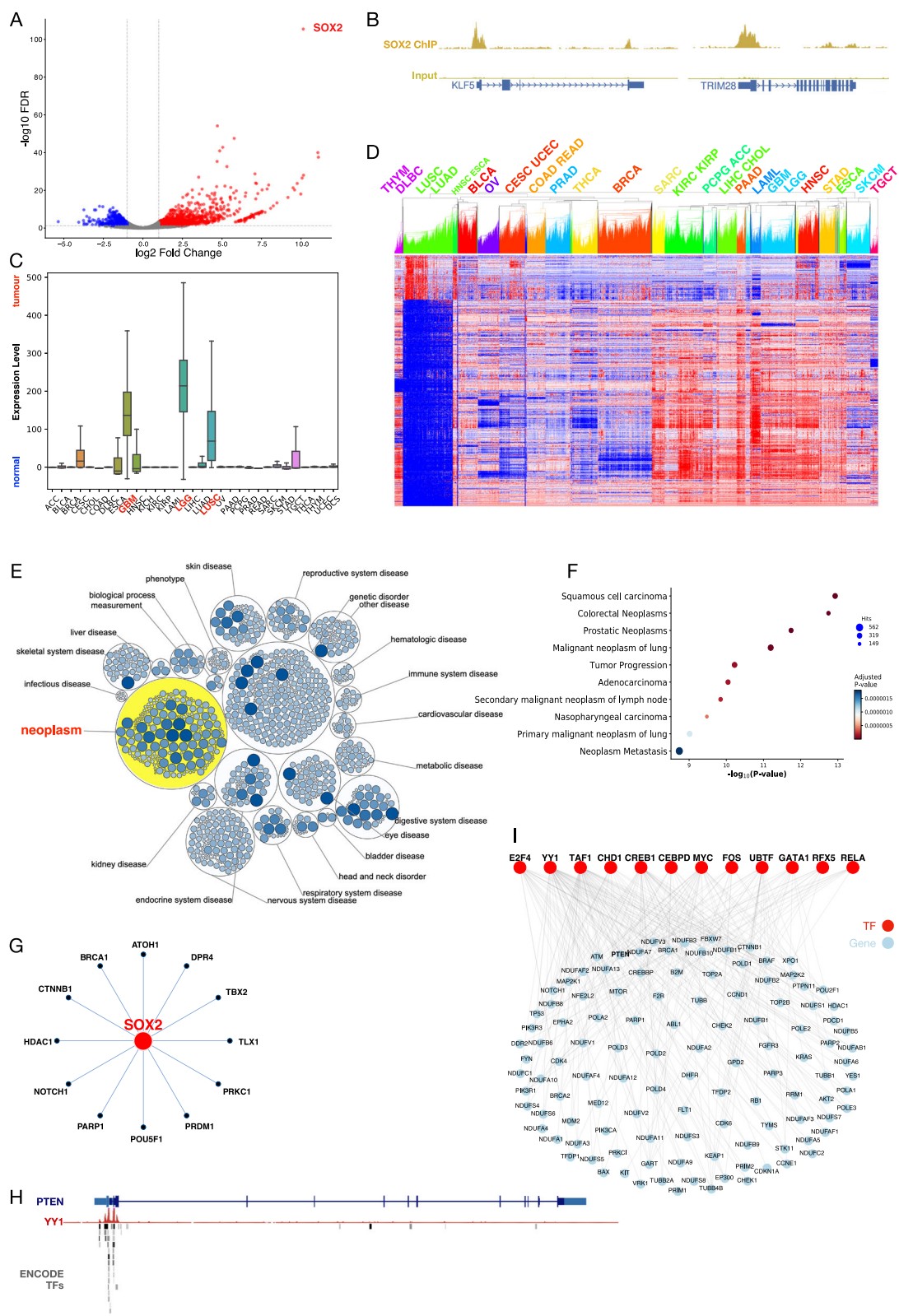

**Communications Chemistry** | (2025)8:401

eigenvector centrality, degree centrality, betweenness centrality, and random walk with restart (RWR). Each algorithm provides unique insights into network dynamics. Eigenvector centrality identifies the most influential proteins by considering both the number of interactions and the importance of interacting proteins. Betweenness centrality identifies proteins that act as bridges between biological pathways. RWR quantifies how information spreads from a target protein to all other proteins in the network, detecting proteins most influenced by the target.

We performed 1,000 permutation tests for each algorithm to identify significantly associated proteins with empirical p-values less than 0.05 (see Methods). Using these methods, we identified 168 key proteins (Fig. 3A, Supplementary Table 3). Fig. 3A provides a Venn diagram showing the

**Fig. 2 | Multi-modal data integration and disease signatures. A** Differential expression analysis from proprietary RNA-seq data identified SOX2 as the most upregulated gene in SOX2-dependent LUSC. **B** SOX2 ChIP-seq analysis revealed direct binding sites of KLF5 and TRIM28, providing insights into its transcriptional regulatory role. **C** Pan-cancer analysis of SOX2 expression in normal versus cancer samples using ENCODE and TCGA data confirmed significant SOX2 upregulation in LUSC. **D** Heatmap of TCGA samples clustered based on LUSC-specific cancer genes identified through comparative analysis of LUSC versus other cancer types. Rows represent individual LUSC-specific genes, and columns represent 31 TCGA cancer types. Red indicates upregulation, and blue indicates downregulation. Lung cancer types (LUSC and LUAD) cluster together, as do other foregut-origin cancers, including squamous types (LUSC, HNSC, and ESCA). **E** Disease enrichment analysis using OpenTargets confirms SOX2's predominant association with neoplasms. **F** DisGeNet over-representation analysis demonstrates that SOX2-dependent DEGs are significantly enriched in epithelial carcinomas, particularly squamous subtypes. **G** Protein-protein interaction (PPI) network from OmniPath shows 12 direct interactors of SOX2. **H** Example of transcription factor binding sites (TFBS) from ENCODE, showing YY1 binding to the PTEN gene. **I** Regulatory map of 12 transcription factors upstream of SOX2, highlighting their interactions with downstream target genes in SOX2-dependent LUSC.

overlap among the proteins identified by the four-centrality metrics. Notably, 143 proteins (85%) were identified by at least two centrality algorithms, demonstrating strong agreement among the methods. Furthermore, 38 proteins were shared across all four algorithms, highlighting their centrality and potential as key targets in the LUSOX network.

Over-representation analysis (ORA) using the DISEASE database revealed that esophageal carcinoma, which is SOX2-associated squamous carcinoma, was the top enriched disease for these key proteins (Fig. 3B). Fig. 3B visualizes the results of the ORA, linking the identified proteins to squamous carcinoma and other related foregut-origin diseases, including esophageal carcinoma and head and neck squamous cell carcinoma. These diseases are also closely related to LUSC in the TCGA cluster (Fig. 2D), further validating the biological relevance of the identified targets. Additionally, the PI3K-AKT pathway, the most altered pathway in LUSC, emerged as the top-enriched biological pathway across all centrality algorithms[24].

To further explore the biological implications, we constructed core networks from the 168 key proteins to investigate their functions and the biological pathways within the core network. Utilizing community detection algorithms, we introduced an innovative approach for pathway identification that diverges from traditional gene enrichment tests. This unbiased method systematically identifies protein communities, corresponding to biological pathways, within a mathematical framework. By reducing reliance on potentially biased pathway databases, this approach offers a more objective means of pathway detection.

Through these community detection algorithms, we identified four distinct communities (Fig. 3C), each associated with specific biological functions: the WNT pathway, PI3K-AKT pathway, cytokine signalling, and cell cycle regulation. Fig. 3C illustrates the core network, with the four identified pathways distinctly colour-coded to highlight their respective clusters. For example, the WNT pathway is shown in purple, cytokine signalling in green, the cell cycle in orange, and the PI3K-AKT pathway in blue. SOX2's central position within the network underscores its pleiotropic influence across these interconnected pathways.

This comprehensive map of interactions and pathways provides a critical resource for identifying and developing treatments that target distinct regulatory mechanisms within the disease.

### Network-based druggable target prioritisation and scoring

To prioritise therapeutic targets for SOX2-dependent LUSC, we utilised 168 key proteins identified through network analysis in the previous section. These key proteins were subsequently used as input for in-silico drug simulations involving 6009 compounds from DrugBank[18], including approved, investigational, and experimental drugs. The simulations identified 521 drugs as potential candidates targeting the key proteins. Fig. 4A illustrates the focused PPI network of drug-targeted proteins, with node size reflecting the number of compounds predicted through in-silico simulation to interact with each protein. We then analysed the subset of 2626 drugs with known indications and identified 217 drugs as potential LUSC therapeutics. Among these, 114 of the predicted drugs overlapped with cancer drugs (of the 2626 drugs, 361 were classified as cancer drugs) (Supplementary Table 3). This overlap was statistically significant (hypergeometric $p$-value = 2.38E-27), supporting the relevance of the identified drug candidates.

**Target Prioritisation.** To further prioritise targets, we performed a structural similarity analysis (Supplementary Fig. 3) to identify clusters of compounds among the 521 drugs, indicating similarity in their target proteins (Supplementary Table 4). Proteins targeted by five or more drugs within each cluster were considered potential primary targets (Fig. 4B, Supplementary Table 5). Fig. 4B provides a detailed dendrogram showing the hierarchical clustering of drugs and their target proteins. The dendrogram reveals that drugs with similar chemical structures tend to target the same key proteins. At the bottom of the dendrogram, the key druggable targets and all upregulated target proteins are highlighted, providing critical insights into the drug-target landscape. Most of the 521 drugs target proteins that are upregulated in SOX2-dependent LUSC, emphasizing their therapeutic potential in addressing this disease context. The alignment between structurally similar drugs and their shared targets further supports the robustness of the prioritisation approach. Of the 168 key proteins, 84 druggable targets were identified as potential primary targets. Among these, 53 showed efficacy in LUSC, as confirmed by DepMap CRISPR results on LUSC cell lines.

### Final prioritisation using iPANDDA scoring.

The 84 druggable targets were further filtered using the iPANDDA scoring function, which resulted in the selection of seven proteins. Three key criteria were applied during this filtering process:

1. Proteins exhibiting downregulated expression in LUSC were excluded.
2. Proteins encoded by genes with a dependency score greater than 0 in LUSC cell lines (based on DepMap data) were excluded.
3. Only proteins belonging to a chemical structure-based drug cluster comprising at least five drugs were retained as druggable targets.

The remaining proteins were then ranked based on two scores:

1. **Proximity score (z-score)**: Measures the proximity of a protein to interact with simulated compounds.
2. **Random Walk Restart (RWR) score**: Quantifies the influence exerted by the SOX2 node on each target protein within the network.

### Pathway analysis of druggable monotherapy targets

The network depicted in Fig. 4C provides a visual representation of the prioritised druggable targets, AKT1, FGFR2, EGFR, VEGFA, PARP1, mTOR, and CDK1, and their connections to SOX2 (Supplementary Table 6). These targets, highlighted as monotherapy candidates, are embedded in the broader network of key proteins influenced by SOX2. The figure emphasizes their central roles in the network and their direct or indirect relationships with SOX2, reinforcing their importance as therapeutic candidates.

From the broader PPI network, we extracted a focused network comprising the 10 prioritised target proteins, including 7 druggable and 3 undruggable targets. Fig. 4C highlights two key upstream druggable pathways for SOX2: the PI3K-AKT and FGFR2 pathways. These pathways are fundamental to cancer progression and treatment response. The PI3K-AKT pathway is frequently deregulated or mutated in LUSC, playing a critical role in cell survival and proliferation. Notably, therapeutic targets within this pathway, such as MTOR and CDK1, demonstrate a complex interplay with transcription factors SOX2 and TP63, as depicted in the network. This

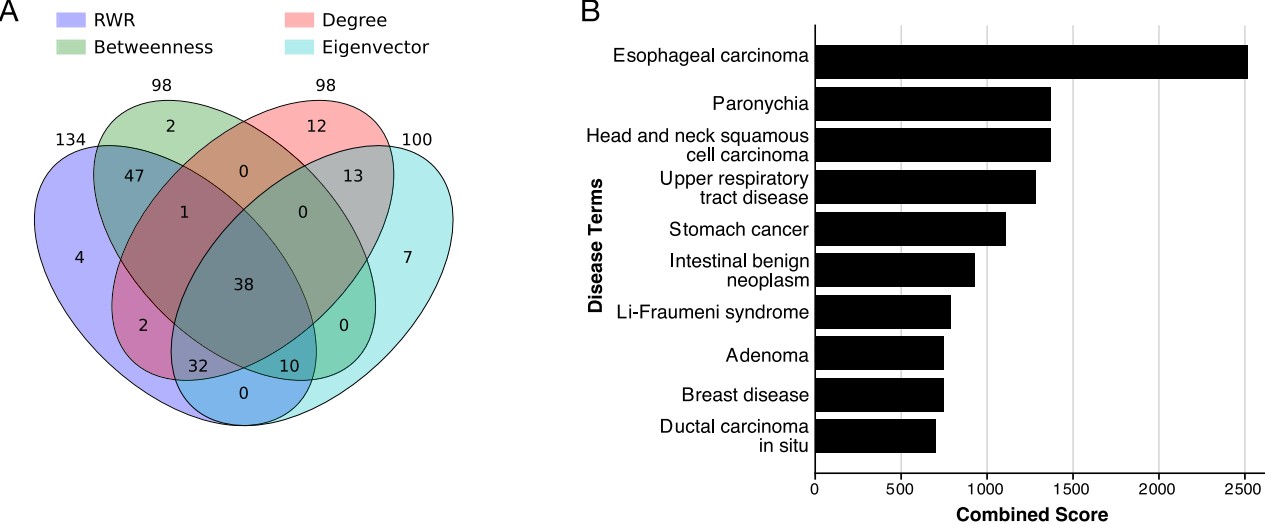

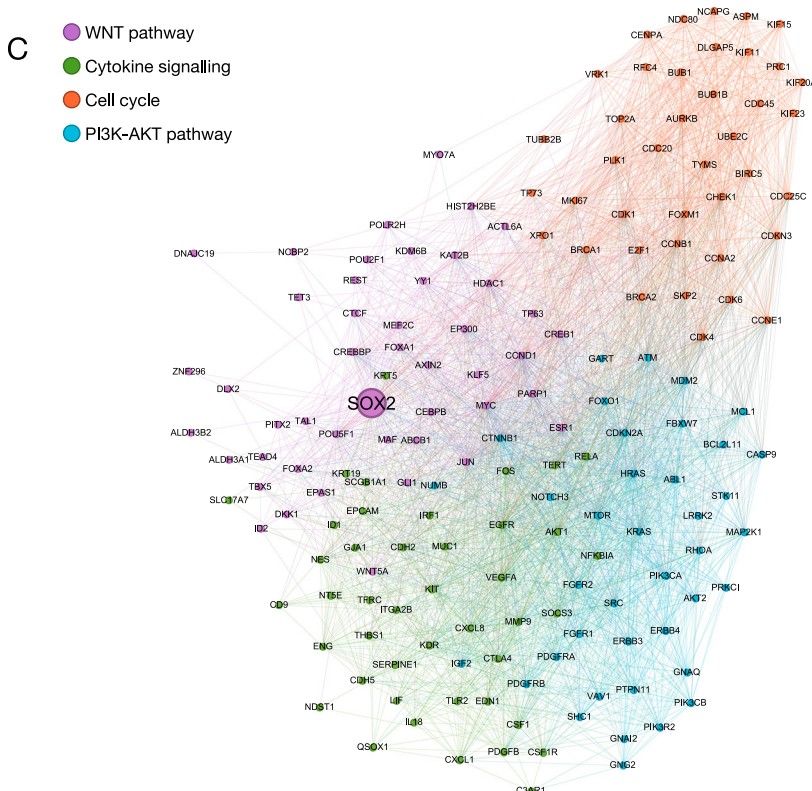

**Fig. 3 | Network-based analysis of the SOX2-dependent LUSC pathways. A** Venn diagram showing the overlap of core proteins identified through network analysis methods: Random Walk with Restart (RWR), Degree centrality, Betweenness centrality, and Eigenvector centrality. A total of 143 proteins (85%) were identified by more than one method, highlighting their critical role in the SOX2-dependent network. **B** Over-representation analysis (ORA) of diseases associated with key proteins in the network, identifying squamous cell carcinoma as the most enriched disease. **C** The core protein network is grouped into four communities, each representing distinct pathways: Cell Cycle, PI3K-AKT Pathway, Cytokine Signalling, and WNT Pathway/DNA Repair. The community containing SOX2 is strongly associated with the WNT Pathway and DNA Repair processes, emphasizing SOX2's central regulatory role in oncogenic pathways.

interplay represents an exploitable vulnerability for therapeutic intervention.

The FGFR2 pathway, another critical axis, is directly associated with SOX2 and represents a promising target for intervention. Both pathways intersect significantly with epithelial-mesenchymal transition (EMT), a key driver of cancer metastasis and recurrence[25,26]. These connections further underscore the strategic importance of targeting these pathways.

This visualisation provides a comprehensive framework for understanding how the prioritised targets align with upstream and downstream regulatory circuits. These insights offer a robust rationale for the selection of these targets for wet-lab validation as monotherapy candidates. Moreover, they lay the groundwork for targeted therapies that can disrupt key regulatory circuits, addressing fundamental mechanisms in SOX2-dependent LUSC pathology.

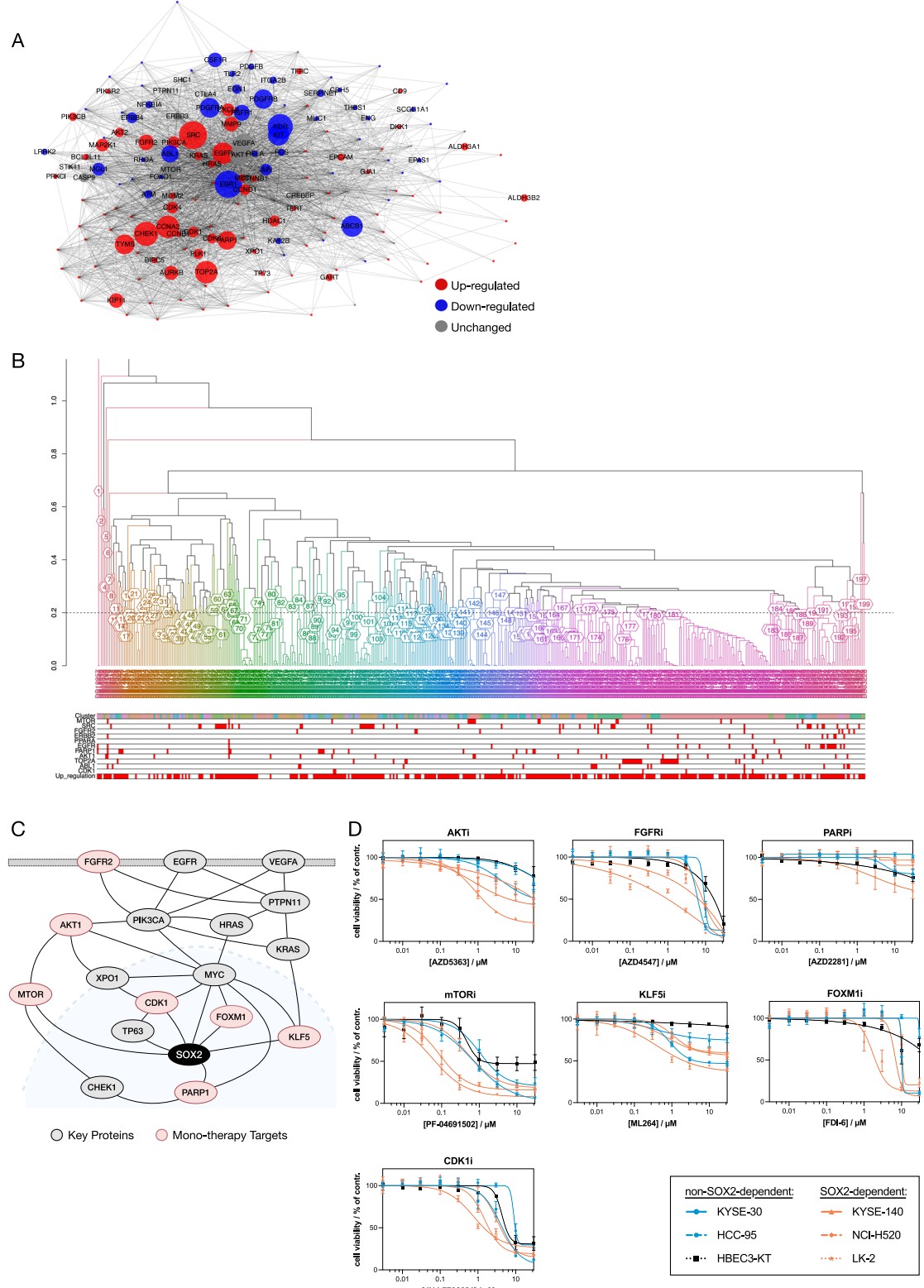

We considered the identified seven druggable targets of monotherapy for experimental testing for efficacy in SOX2-dependent squamous cell lines. Two of the target proteins, EGFR and VEGFA, from our predicted list have already been tested in clinical trials and were therefore not retested in an in vitro context. EGFR inhibition has been shown to be ineffective in LUSC[27,28], but targeting VEGF using a monoclonal antibody in combination with PD1 has recently been demonstrated as effective in a landmark study[29]. Notably, prior clinical complications, specifically fatal pulmonary haemorrhage, rather than a lack of efficacy, precluded the use of VEGF-targeting therapies in LUSC[30]. It is noteworthy that iPANDDA identified targets with a sufficient rationale for investment in clinical trials, reaffirming the validity of the computational prioritisation.

**Fig. 4 | Prioritisation, drug simulation, and experimental validation of druggable monotherapy targets for SOX2-dependent LUSC. A** Interaction network of candidate targets. Red nodes represent upregulated proteins, and blue nodes represent downregulated proteins in LUSC. Node size corresponds to the number of candidate drugs targeting each protein, with larger nodes such as MTOR and CDK1 indicating highly targeted hubs. **B** Hierarchical clustering dendrogram illustrating the relationships between candidate drugs. The colours indicate pathway classifications, such as the PI3K-AKT pathway, cell cycle regulation, and WNT signalling. Upregulated targets are annotated at the bottom, aligning with specific drug clusters. This panel highlights the overrepresentation of upregulated targets in key oncogenic pathways, further supporting their therapeutic potential in SOX2-dependent LUSC.

**C** Druggable pathways associated with the prioritised monotherapy targets. The network highlights connections between FGFR2, PI3K-AKT, and RAS pathways as upstream regulators of SOX2, with implications for therapeutic intervention. **D** Dose-response curves showing the effect of inhibitors targeting the 7 prioritised monotherapy proteins in SOX2-dependent (orange) and non-SOX2-dependent (blue, black) cell lines. Cells were treated with half-logarithmic concentrations from 3 to 30,000 nM for 72 h, and viability was assessed using a resazurin-based assay. Data are presented as mean ± SEM from three independent experiments. The distinct sensitivity of SOX2-dependent cell lines, particularly to inhibitors such as PF-04691502 (MTOR inhibitor) and JNJ-7706621 (CDK1 inhibitor), demonstrates the specificity and efficacy of these prioritised targets.

For the remaining prioritised targets, AKT1, mTOR, CDK1, FGFR2, and PARP1, and to quantify the efficacy of drugs targeting these proteins, we established a panel of SOX2-dependent and non-SOX2-dependent squamous cell lines (Supplementary Fig. 4A). We performed a series of dose-response cell population expansion assays using half-logarithmic concentrations ranging from 3 to 30,000 nM for 72 h.

Importantly, as shown in Fig. 4D, there was a clear distinction in the sensitivity of SOX2-dependent cell lines compared to non-SOX2-dependent cell lines when treated with inhibitors targeting AKT1 (AZD5363), mTOR (PF-04691502), and CDK1 (JNJ-7706621). The dose-response curves illustrate a significantly enhanced efficacy of these compounds in SOX2-dependent cell lines, reinforcing their potential as candidate therapeutics for SOX2-dependent LUSC. Figure 4D specifically highlights the steep decline in cell viability for SOX2-dependent cell lines compared to non-SOX2-dependent ones, underscoring the specificity of these inhibitors. This differential response provides robust evidence supporting iPANDDA's capability to identify potential therapeutic approaches tailored for SOX2-dependent LUSC. Given PF-04691502 targets both MTOR and PI3K, further experiments using two distinct MTOR inhibitors confirmed that the differential response was a group effect in response to MTOR inhibition (Supplementary Fig. 4B)

It is important to note that iPANDDA does not imply directionality of the impact of targeting proteins. For instance, while PARP1 was identified as a target protein, both our experimental results (Supplementary Fig. 4C) and data from therapeutic screening databases (Supplementary Fig. 5) indicate that PARP inhibition may paradoxically potentiate squamous cancers. This observation aligns with a recent study reporting the failure of PARP inhibition as part of a second-line treatment algorithm for metastatic squamous lung cancer[31].

## Prioritisation and validation of non-druggable targets and combination therapies

For the remaining 84 proteins not currently targeted by drugs, we performed a similar ranking analysis. For these target proteins lacking known targeting drugs from our simulations, we applied a multi-step filtering process followed by prioritization:

1. Proteins downregulated in lung squamous cell carcinoma (LUSC) samples from the TCGA database were excluded.
2. Proteins with a gene dependency score greater than 0 in LUSC cell lines were filtered out based on data from the DepMap resource.
3. Proteins not directly interacting with the SOX2 transcription factor in our LUSC-specific network (LUSOX) were removed.

After this three-step filtering process, the remaining proteins were prioritized based on their random walk restart (RWR) score from the SOX2 protein node in the network. For non-druggable targets, compound–target interaction data were not available, and thus network proximity could not be assessed; therefore, prioritization was based solely on the RWR score. Three non-druggable target proteins, p63, FOXM1, and KLF5, met the threshold for experimental validation (Supplementary Fig. 4C–E, Supplementary Table 6). FOXM1 and KLF5 are potentially important transcription factors in LUSC. FOXM1 has been implicated in various hallmarks of cancer, including cell cycle progression and

regulation of the tumour microenvironment as well as in LUSC progression and in the regulation of SOX2 transcription[32–35]. Functionally, KLF5 promotes cell proliferation, migration, cell cycle progression, and is anti-apoptotic[36]. As a transcription factor, KLF5 increases the expression of proteins involved in cell cycle regulation (Cyclin D1, Cyclin B1, p27), angiogenesis (VEGFA), apoptosis (Survivin), and stemness (Nanog)[37]. KLF5 interacts with SOX2 and p63 to maintain chromatin accessibility in oesophageal squamous carcinoma, and these transcription factors coregulate each other[38]. Moreover, multiple mechanisms of KLF5 deregulation have been implicated in the pathogenesis of head and neck squamous carcinoma.

The effective treatment of cancer increasingly involves combination therapies. Leveraging our network analysis, we explored whether previously unexplored therapeutic approaches could be developed by combining one of these non-druggable targets with a druggable target. We previously grouped the SOX2-related LUSC pathways into four functional communities: cell cycle, cytokine signalling, PI3K-AKT pathway, and WNT pathway (Fig. 3C). Based on network medicine principles suggesting that drug combinations targeting proteins located in distinct network communities achieve enhanced therapeutic efficacy through complementary perturbation of disease pathways and a reduced likelihood of adverse drug effects, we selected combination targets from separate functional communities.

Within the WNT pathway community, we identified the transcription factor TRIM28 as a critical regulator, using transcription factor enrichment analysis with the ENCODE TF binding site dataset (see Methods: Identification of combination therapy). TRIM28 was the most significantly enriched transcription factor, influencing eight proteins that interact with SOX2 (Fig. 5A). Given its central regulatory role within the WNT-SOX2 network, TRIM28 was prioritized as a candidate for combinatorial targeting strategies. We evaluated the combinatorial efficacy of TRIM28 knockout with inhibition of key druggable proteins from other communities, namely CDK1 (cell cycle), AKT (cytokine signalling), and FGFR (PI3K-AKT pathway) (Fig. 5B). This cross-community targeting approach aimed to achieve synergistic therapeutic effects through coordinated perturbation of complementary signalling pathways.

Figure 5C highlights the results of this combination therapy strategy. Notably, in the LK-2 cell line, which exhibits elevated expression of both TRIM28 and SOX2 alongside strong SOX2 dependency, there were significant synergistic responses when TRIM28 knockout was combined with inhibitors of AKT, FGFR, or CDK1. This strong synergy in LK-2 cells underscores the biological relevance of TRIM28 as a potential combination target (Supplementary Fig. 6). Fig. 5D provides additional evidence for the importance of TRIM28 in SOX2-dependent LUSC. It shows the relationship between TRIM28 expression and SOX2 expression across cell lines, as well as their dependency scores derived from CRISPR knockout screens. The elevated TRIM28 and SOX2 expression in the LK-2 cell line correlates with the strong combinatorial effect observed in experimental assays, further validating the selection of TRIM28 as a critical node for combination therapies.

TRIM28 is not a currently druggable target, but this work demonstrates that by separating targets into functional communities, iPANDDA can uncover previously unexplored approaches to combination therapy. The

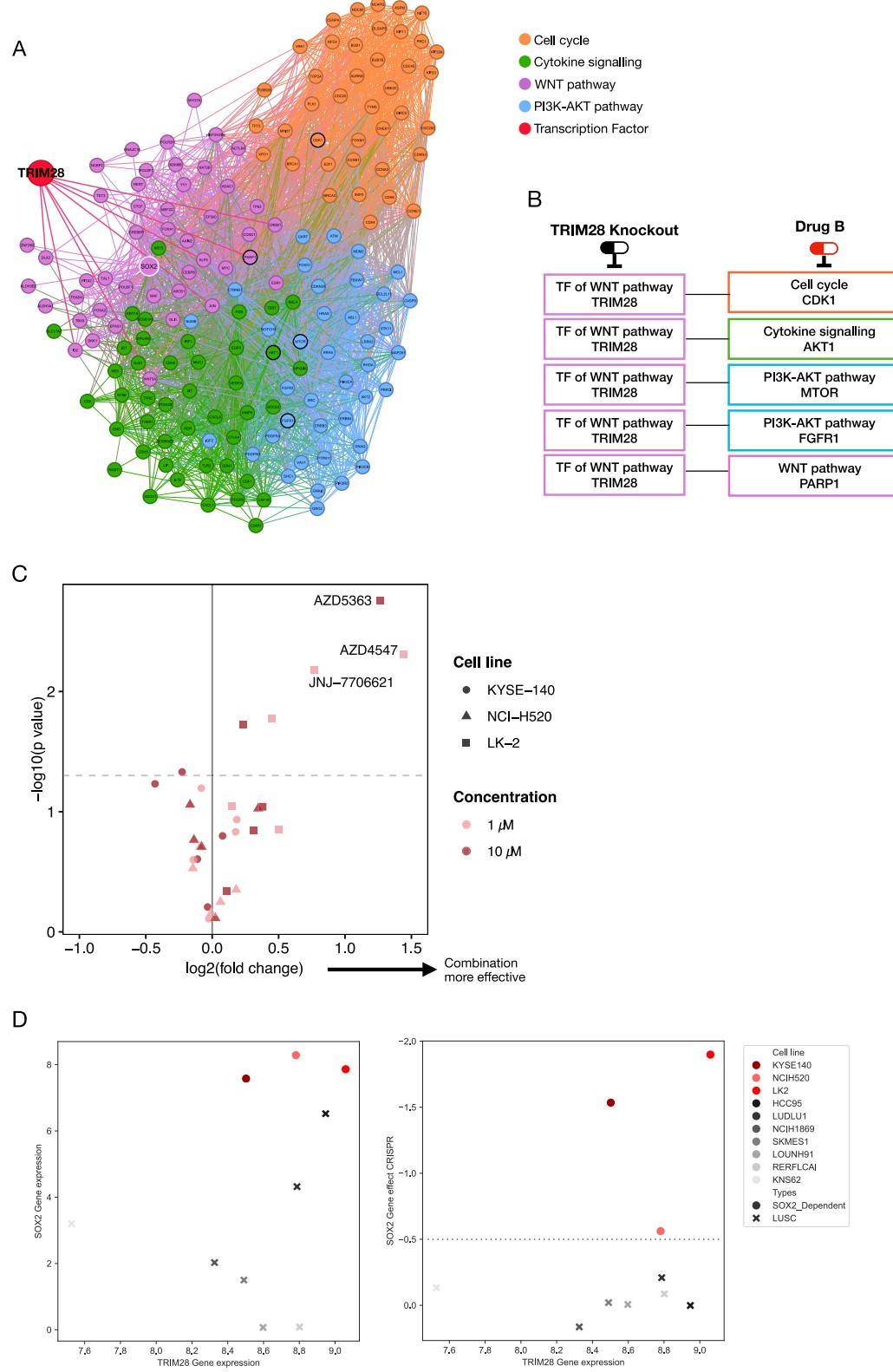

ability of iPANDDA to recognize and prioritize targets based on their position within distinct network communities, while maintaining focus on SOX2 dependency, highlights its sophisticated understanding of disease-specific biological systems. This capability is particularly critical for LUSC, where the pleiotropic effects of SOX2 suggest that combination therapies may be necessary for effective treatment.

## Discussion

This study introduces iPANDDA, a computational pipeline developed to identify and prioritise therapeutic targets for both monotherapy and combination therapy in SOX2-dependent lung squamous cell carcinoma (LUSC). By integrating and analysing multi-modal datasets, iPANDDA established a robust disease-specific protein-protein interaction (PPI)

**Fig. 5 | Prioritisation, network analysis, and experimental validation of non-druggable combination therapy targets for SOX2-dependent LUSC. A**) TRIM28 was identified as a significantly enriched transcription factor (TF) within the WNT pathway community using a TF enrichment analysis of core network communities. TRIM28 directly regulates eight proteins in the core network, highlighting its critical role in the regulatory landscape of LUSC. **B** Proposed combination therapy strategies involve TRIM28 knockout paired with drugs targeting proteins from the four communities of the iPANDDA prioritised protein network: cell cycle, cytokine signalling, PI3K-AKT pathway, and WNT pathway. **C** A volcano plot depicts the effects of drug treatment targeting AKT, CDK1, FGFR, mTOR, and PARP in combination with TRIM28 knockout. Experiments were performed in two SOX2-dependent cell lines (LK-2 and KYSE-140). Cells were engineered using guide RNAs (sgRNA) for TRIM28 knockout or a non-targeting control sgRNA, and treatments were applied at 1 µM or 10 µM for 72 h. Cell viability was assessed, and compounds in the top right quadrant of the plot demonstrate significant enhancement of efficacy in combination with TRIM28 knockout. The full list of drug names is provided in Supplementary Fig. 6C for reference **D** The LK-2 cell line exhibited the strongest response to combination therapy (**C**). Elevated expression levels of both TRIM28 and SOX2 in LK-2 cells, along with a pronounced SOX2 gene dependency effect, were observed compared to other LUSC cell lines. This highlights the importance of TRIM28 and SOX2 co-dependency in identifying therapeutic vulnerabilities specific to SOX2-dependent LUSC.

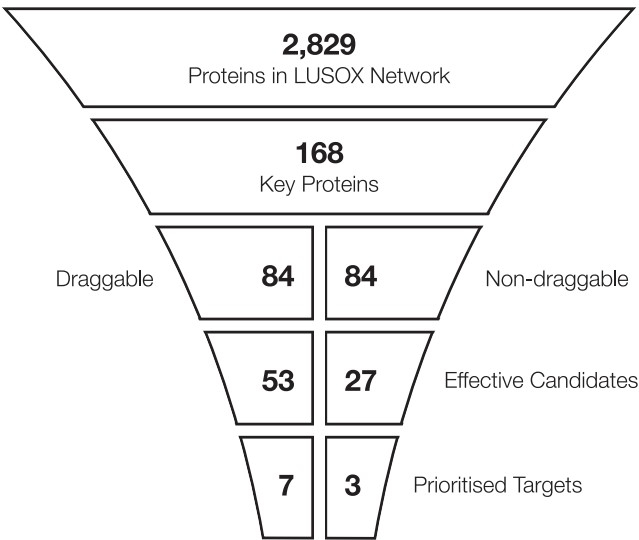

**Fig. 6 | Stepwise prioritisation of druggable and non-druggable targets in SOX2-dependent LUSC.** A systematic pipeline from the LUSOX network (2,829 proteins) identified 168 key proteins through network analysis. Drug simulations stratified these into druggable ($n = 84$) and non-druggable ($n = 84$) proteins. Parallel prioritisation approaches refined druggable proteins to 53 effective candidates and non-druggable proteins to 27, further narrowed to seven druggable and three non-druggable prioritised targets for monotherapy.

network, identifying 168 core proteins implicated in LUSC pathogenesis. The stepwise prioritisation of these proteins through drug simulations and network algorithms revealed seven druggable and three non-druggable monotherapy candidates (Fig. 6).

Fig. 6 summarises the stepwise prioritisation process, starting with 2829 proteins in the LUSOX network. Through systematic network analysis, 168 key proteins were identified based on centrality metrics and biological relevance. Drug simulations were then performed using 6009 compounds from DrugBank to evaluate their potential interactions with the 168 key proteins, resulting in 521 candidate drugs targeting these proteins. Based on the simulation results, the 168 key proteins were stratified into two groups: druggable ($n = 84$) and non-druggable ($n = 84$).

From this point, a parallel approach was taken to further refine the druggable and non-druggable groups. For the druggable proteins, dependency scores, expression levels, and clustering within drug-target interaction networks reduced the pool from 84 to 53 proteins. Simultaneously, the candidate drugs were refined from 521 to 217 with potential LUSC therapeutic relevance. Among these, 114 drugs overlapped with cancer drugs, underscoring their biological and therapeutic significance. For the non-druggable proteins, similar dependency scores and network analyses reduced the pool to 27 effective candidates.

Further prioritisation narrowed the druggable proteins to seven high-priority targets based on expert reviews, experimental feasibility, and clustering analyses. In parallel, the non-druggable proteins were refined to three prioritised targets using combinatorial therapy assessments (Fig. 5A, B),

alongside expert reviews and feasibility considerations. Experimental validation confirmed the efficacy of inhibitors targeting AKT1, MTOR, and CDK1 for monotherapy in SOX2-dependent LUSC cell lines, while combinatorial evaluations demonstrated the relevance of non-druggable targets for combination therapies. These results highlight the systematic and iterative nature of iPANDDA in narrowing down a large protein network to actionable targets and therapeutic candidates. SOX2 is not directly targeted by any of these compounds - rather the network analysis has suggested these druggable targets may confer vulnerability when dependency on this pleiotropic and promiscuous transcription factor is present.

In addition to monotherapy targets, iPANDDA identified TRIM28 as a critical regulator within the SOX2-centric network. TRIM28 emerged as a promising combination therapy target, particularly when paired with inhibitors of AKT, CDK1, or FGFR1 from other core pathway communities. Experimental evaluations demonstrated selective effectiveness of these combinations in SOX2-dependent LUSC models, particularly in the LK-2 cell line, which exhibits high SOX2 and TRIM28 dependency. These findings highlight iPANDDA's capacity to address the molecular complexities of LUSC and identify actionable opportunities for combination therapies.

The integration of diverse datasets and systematic prioritisation provided a comprehensive framework for identifying both well-characterised and potential targets. This approach significantly expands the therapeutic landscape for SOX2-dependent LUSC. The in vitro validation experiments further emphasised the importance of tumour molecular characteristics—specifically SOX2 dependency—in determining therapeutic response. The identification of TRIM28 as a combination therapy target illustrates the transferability of iPANDDA's methodology, offering a template for similar analyses in other cancers.

However, certain limitations should be acknowledged. The predictive power of iPANDDA relies on the completeness and accuracy of protein interaction and drug target databases. Additionally, current in vitro models may not fully replicate the complexity and heterogeneity of the human tumour microenvironment, necessitating complementary in vivo studies to validate findings further. Further, due to the unexplained challenges in establishing LUSC cell lines that robustly express the canonical markers of LUSC - KRT5, p63 (p40), and SOX2, we utilised a squamous oesophageal cell line that retains these canonical markers. This reflects a growing trend for tissue-agnostic cancer therapeutics[39].

Despite these limitations, iPANDDA represents an innovative and comprehensive approach that integrates multi-modal datasets within a unified analytical framework, offering a systematic solution to therapeutic target discovery. While the current study focused on SOX2-dependent LUSC, this methodology is broadly applicable to other cancers, enabling the identification of previously unexplored targets across a wide range of disease contexts. Future efforts will aim to incorporate machine learning techniques to refine target prioritisation, particularly for mono- and combination therapies, further enhancing the pipeline's accuracy and translational potential.

## Methods
### Chromatin immunoprecipitation (ChIP)
Cells cultured in OTC and untreated/treated with 2 µg/ml Dox for 4 days were harvested. To harvest cells cultured in the OTC, medium was removed

from both lower and upper chambers and the porous membrane of each transwell was wrapped in cling film to ensure the cells were fully submerged into the solutions. The collagen disks were washed twice with 500 μL of Phosphate-Buffered Saline (PBS-Sigma) at RT and trypsinised with 3x trypsin-EDTA for 12 min at 37 °C. Trypsinised cells from 3 transwells (under the same experimental condition) were collected into micro-centrifuge tubes containing DMEM/FBS and centrifuged at 1100 rpm. Approximately $1 \times 10^6$ cells were used for ChIP assay, carried out according to the manufacturer's instructions (Magna ChIP™ G, Millipore, Cat No. 17-611). Briefly, cells from 5 different transwells were collected into the same tube and cross-linked by resuspending the cell pellet in 10 ml of KSFM +supplements with 275 μL of 37% formaldehyde for 10 min at RT. The reaction was quenched by addition of 1 ml of 10x glycine and washed twice with cold 1x PBS. The fixed cells were resuspended in cell lysis buffer. Nuclei were collected by centrifugation at 2000rcf and resuspended in nuclear lysis buffer. Samples were then sonicated to produce chromatin fragments of an average of 500 bp. Chromatin shearing was performed in 130 μL microtubes (with 130 μL total volume) using the sonicator Covaris S2 series and the following programme: Duty cycle: 2%. Intensity: 3. Cycles per burst: 200. Duration: 12 min.

50 μL of sheared chromatin was incubated with 20 μL protein G magnetic beads 5 μg of either Goat anti-SOX2 antibody (AF2018) or Goat anti-IgG control antibody (AB-108) overnight at 4 °C with rotation. Next day, the protein–DNA complex was washed, the cross-link was reversed, and proteins were removed by proteinase K treatment. DNA was purified using spin columns and finally eluted in 50 μL of elution buffer.

For the validation of immunoprecipitation, the sheared chromatin was incubated with a validated antibody and a negative control, followed by PCR of the eluted DNA (ChIPAb+ RNA Pol II – ChIP Validated Antibody and Primer set, Millipore, Cat No. 17-672).

Initial sample quality control of pre-fragmented DNA was performed using a Tapestation DNA 1000 High sensitivity Screen tape (Agilent, Cheadle UK). Sequencing-ready libraries were prepared from samples using the Hyper Prep DNA Library preparation kit (Kapa Biosystems, London UK) and indexed for pooling using NextFlex DNA barcoded adapters (Bioo Scientific, Austin TX US). Libraries were quantified on a Tapestation DNA 1000 Screen tape and by qPCR using an NGS

Library Quantification Kit (KAPA Biosystems) on an AriaMx qPCR system (Agilent). Libraries were then normalised, pooled, diluted and denatured for sequencing on the NextSeq 500 (Illumina, Chesterford UK). Sequencing was performed using a high output flow cell with 2 × 75 cycles of sequencing.

## ChIP-seq data processing
Raw sequencing reads were assessed for quality using FastQC. High-quality reads were aligned to the human reference genome (GRCh38) using Bowtie2, and alignment statistics were analysed using Samtools. Aligned BAM files were sorted and indexed with Samtools, and duplicate reads were marked and removed using Picard Tools to minimize artifacts. Normalized coverage tracks (bigWig files) were generated using bamCoverage from the deepTools suite to visualize genome-wide SOX2 binding profiles. Signal differences between ChIP samples and controls were quantified using bamCompare from deepTools, enabling the generation of differential signal profiles.

## Multi-modal signatures
Transcriptomics disease signatures (TCGA & GTEx): Lung squamous cell carcinoma (LUSC)-specific cancer genes were identified by analysing transcript per million (TPM) values from the LUSC-TCGA TARGET GTEx dataset, a combined cohort of TCGA, TARGET, and GTEx data provided by UCSC Xena[12]. The full dataset comprises 17,221 samples, including 9807 from TCGA and 7414 from GTEx. For LUSC, the data comprise 498 disease samples (TCGA) and 337 healthy samples, including 50 from TCGA and 287 from GTEx. For each disease sample, the fold-change was calculated as the ratio to the average of healthy samples (TCGA + GTEx). Subsequently,

genes were filtered using an ANOVA test to identify those exhibiting significant differential expression in LUSC compared to other cancer types in the TCGA pan-cancer dataset (31 cancer types) and matched normal tissues from the GTEx dataset (30 tissue types). The genes were ranked based on their ANOVA p-values from these comparisons. A distinct sharp increase in p-values was observed after the threshold of 1e-150. Therefore, genes with p-values ≤ 1e-150 were selected as the LUSC-specific gene set, applying a highly stringent cutoff to ensure high specificity.

Using this LUSC-specific gene set, hierarchical clustering was performed on samples from 31 cancer types present in the TCGA database to explore cancer type-specific expression patterns and potential subgroups or subtypes based on the expression profiles of these genes.

LUSC functional signatures: LUSC-associated genes were selected from the Open Targets 19.09[14]. The Open Targets platform integrates evidence from various data sources, including genetics, genomics, transcriptomics, drug information, animal models, and scientific literature, to compute association scores between targets and diseases. Genes with an overall association score of >= 0.8 were considered as LUSC-associated genes. We chose a threshold value of 0.8 for inclusion based on a disease enrichment analysis (Enrichr[40]) performed on gene lists with a range of overall association score thresholds (0.75, 0.8, 0.85, 0.9). The threshold of 0.8 exhibited the lowest p-value for enrichment in squamous cell carcinoma of the lung, indicating the most significant association with the disease of interest.

Drug Target Signatures: Drug target signatures consist of proteins that are established as targets of drugs currently in Phase 3 or higher clinical trials for lung squamous cell carcinoma (LUSC). The data on LUSC drugs and their corresponding protein targets were sourced from the Open Targets 19.09 database.

## LUSOX network construction
The LUSOX network was constructed using a multi-modal signature PPI network from the STRING database. Only interactions with a confidence score of greater than medium (0.4) were used. Networks were visualized using Gephi 0.9.2[41] (Supplementary Fig. 2A).

## LUSOX network community detection
The Louvain community detection algorithm, implemented in the Gephi software, was applied to LUSOX to identify communities. The Louvain method is a hierarchical clustering approach that optimizes modularity to detect communities of nodes that are more densely connected within themselves compared to connections between communities.

## Network analysis to identify key proteins
Eigenvector centrality, degree centrality, betweenness centrality, and random walk with restart (RWR) were utilized to identify key proteins in the LUSOX network. The LUSOX network was represented by an adjacency matrix $A$, where $A_{ij} = 1$ there is an edge between nodes $i$ and $j$ or $A_{ij} = 0$ otherwise. The eigenvector centrality $x_i$ was defined as

$$\lambda x = xA \tag{1}$$

where $x$ is an eigenvector of the adjacency matrix $A$ with eigenvalue $\lambda$. If $\lambda$ is the largest eigenvalue of the adjacency matrix $A$, there is a unique solution $x$, all centrality values are positive[42]. Degree centrality of node $i$ was defined as

$$C_D(i) = \sum_{j=1}^{N} A_{ij} \tag{2}$$

where $N$ is the number of nodes in the LUSOX network. Betweenness centrality of a node $i$ was defined as

$$C_B(i) = \sum_{s,t \in V} \frac{\sigma(s,t|i)}{\sigma(s,t)} \tag{3}$$

where $V$ is the set of nodes, $\sigma(s, t)$ is the total number of shortest paths between $s$ and $t$, and $\sigma(s, t|i)$ is the number of shortest paths between $s$ and $t$ paths passing through node $i$. If $s = t$, $\sigma(s, t) = 1$, and if $i \in s, t, \sigma(s, t, |, i) = 0$.

Eigenvector centrality was used to identify the most influential proteins in the network. If a protein frequently interacts with other proteins that also have high eigenvector centrality, then the protein will have high eigenvector centrality. Degree centrality was used to identify the hub proteins in the network. Betweenness centrality was used to identify the bottleneck proteins in the network. A RWR algorithm was used to see which human proteins are most affected by SOX2 protein. To do this, we used SOX2 as the starting point of RWR. The RWR parameters were (1) a restart probability of 0.15, (2) a maximum iteration number of 100, and (3) an error tolerance of 1e-06. We assigned edge betweenness centrality as an edge score on the LUSOX. The RWR calculated a score per protein in the LUSOX network, which indicates how much of a given protein was influenced by SOX2. The algorithms were implemented in the Python package NetworkX (v 2.2)[43].

Permutation tests were performed 1,000 times to identify significant proteins for each of the network centrality algorithms. In 1,000 permutation tests, each test generated a random network with a preserved degree distribution of the original network, the LUSOX network. To generate a random network, we reconnected the edge in the LUSOX network and swiped the node. So, the random network in each permutation test has at least 66% of the rewired edges. Then, in the permutation test, we applied the network algorithm and obtained the cumulative results of the network algorithm. These cumulative results were used to calculate the empirical p-value of the network algorithm. We combined the four permutation test results to determine the final set of key proteins that have an empirical p-value <=0.01 in either result.

### Drug-Target interactions
Approved drugs were collected from ChEMBL[44] and DrugBank[18]. Drug-target interaction information was collected from DrugBank(v 5.1)[18], STITCH (confidence score > 0.9)[45] and Cheng, et al[46].

### Drug simulation
Approved drugs were collected from ChEMBL[44] and DrugBank[18]. Drug-target interaction information was collected from DrugBank (v 5.1)[18], STITCH (v 5.0, confidence score > 0.9)[45] and Cheng et al.[46].

In-silico network-based proximity analysis was conducted for key proteins from the LUSOX network. Given K, the set of key proteins from LUSOX networks, and T, the set of drug targets, the network proximity(Eq. (5)) of K with the target set of T of each approved drug where d(k, t) is the shortest path length between nodes k ∈ K and t ∈ T in the human PPIs[46] was executed. Closest distance measure was used to calculate the distance between a given drug's targets and key proteins in the LUSOX network, because it previously showed best performance in drug-disease pair prediction in the study of Guney et al.[47],.

$$d_c(K, T) = \frac{1}{||T||} \sum_{t \epsilon T} \min_{k \epsilon K} d(k, t) \quad (5)$$

To assess the significance of the distance between a key protein in the LUSOX network and a drug $d_c(K, T)$, the distance was converted to Z-score based on permutation tests by using

$$z(K, T) = \frac{d(K, T) - \mu_{d(K,T)}}{\sigma_{d(K,T)}} \quad (6)$$

The permutation tests were repeated 1000 times, each time with two randomly selected gene sets. There are few high-degree nodes due to the scale-free network of the human protein–protein interaction network. To avoid repetitive selection of the same high-degree nodes during random selection, we used a binning approach with at least 100 nodes in a bin. In the binning approach, nodes in same bin have a similar node degree to maintain

node degree distribution for random selection. When we randomly select a set of genes, we perform a random selection among proteins from all bins to verify that the minimum node degree was less than the minimum node degree of the selected gene set and the maximum node degree was greater than the maximum node degree of the selected gene set. The corresponding p-value was calculated based on the permutation test results. Drug to SOX2-dependent LUSC associations with a Z-score of less than −2 were considered significantly proximal[47].

### Prioritisation of key proteins for therapy
Three key criteria are employed to filter potential protein targets: 1) Proteins exhibiting downregulated expression in lung squamous cell carcinoma (LUSC) samples from the TCGA database are excluded, as the majority of compounds in our in-silico drug screening act as inhibitors, which are more effective against upregulated proteins. (Supplementary Fig. 7) 2) Proteins encoded by genes with a dependency score greater than 0 in LUSC cell lines, based on data from the DepMap database[16], are filtered out. 3) Only proteins belonging to a chemical structure-based drug cluster comprising at least five drugs are considered as druggable targets.

The remaining proteins after this triple-filtering process are then prioritized based on two scores:

$$Monotherapy\ score = z \times 1/\log(RWR)$$

i) Proximity score (z-score) from drug simulations
ii) Random walk restart (RWR) score from the SOX2 protein

The proximity score calculated the proximity of a protein to interact with the compounds simulated, while the RWR score quantifies the influence exerted by the SOX2 node on each target protein within a network analysis framework. Thus, targets are ranked according to their predicted druggability (proximity score) and functional relationship with the SOX2 TF (RWR score).

For proteins without compound–target interactions—and thus no drugs predicted to target them in our drug simulation (i.e., non-druggable targets)—we applied a different multi-step filtering and prioritization process: (1) Proteins downregulated in lung squamous cell carcinoma (LUSC) samples from the TCGA database were excluded. (2) Proteins with a gene dependency score greater than 0 in LUSC cell lines were filtered out based on data from the DepMap resource. (3) Proteins not directly interacting with the SOX2 TF in our LUSC-specific network (LUSOX) were removed.

After this three-step filtering process, the remaining proteins were prioritized based on their random walk restart (RWR) score from the SOX2 protein node in the network. For non-druggable targets, compound–target interaction data were not available, and thus network proximity could not be assessed; therefore, prioritization was based solely on the RWR score.

### Key protein network community detection
Key protein network is derived from the LUSOX network, which consists of 168 proteins. The Louvain community detection algorithm, implemented in the Gephi software, was applied to this key protein network to identify communise.

### Identification of combination therapy
Combination drug targets were selected based on the following criteria: (1) The pair of targets was chosen such that each drug targets a different community. (2) One target was selected as the TF (TRIM28) that regulates the entire community containing SOX2. (3) The other target was selected from the list of monotherapy candidates, with each candidate belonging to a different community within the network.

TF enrichment analysis was performed on the community/module containing the SOX2 protein, identified as the WNT pathway community. The enrichment was evaluated using the Enrichr and the ENCODE TF

binding site dataset. An over-representation test was conducted, and TRIM28 was identified as this community's top enriched TF.

## Cell viability assay
Cells were incubated for 24 h before adding drugs in DMSO in half-logarithmic concentrations from 3 to 30,000 nM. After incubation for 72 h, the alamarBlue assay was used to assess cell viability by measuring fluorescence intensity. Dose-response curves were fitted in Graphpad Prism using data points from at least three biological replicates.

## In-vitro validation of combination therapies
PX458 was a gift from Feng Zhang (Addgene # 48138). Previously validated sgRNA sequences targeting TRIM28 and a non-targeting control were cloned into PX458 to express sgRNAs and Cas9, and transfected into cell lines using lipofectamine LTX. After 72 h, GFP-expressing cells were sorted on the Aria III cell sorter, expanded and used in cell viability assays to assess combinatorial effects.

## Phenotyping of cell line markers
Cells were harvested and protein lysates immunoblotted as described in Correia et al. 2017[48]. RNA was harvested using the RNeasy Mini kit, cDNA synthesised with the High-Capacity cDNA Reverse Transcription kit, and RT-qPCR performed with SYBR Green reagents. Data was analysed using the ΔΔCt method and were normalised to TBP expression.

## Data availability
The datasets generated during and/or analysed during the current study are available in the Zenodo repository (https://doi.org/10.5281/zenodo.17143330).

## Code availability
All code generated in this study is openly available on Zenodo at https://doi.org/10.5281/zenodo.17143330.

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

## Acknowledgements

W.H., M.M., and N.H. were funded by LifeArc. D.K. was supported by the MRC DTP at the University of Cambridge. L.C. and F.M. were funded by the Wellcome Trust (WT097143MA). F.M. and W.G. were supported by the BBSRC (BB/W014564/1) and the NC3Rs (NC/S001204/1). This research was supported by the National Institute for Health, Care Research (NIHR) Cambridge Biomedical Research Centre (NIHR203312), the Korea Planning & Evaluation of Industrial Technology funded by the Ministry of Trade, Industry and Energy (RS-2024-00410585), the National Research Foundation of Korea grant funded by the Ministry of Science and ICT (RS-2025-18362970), a grant from Korean ARPA-H Project through the Korea Health Industry Development Institute (KHIDI) funded by the Ministry of Health & Welfare, Republic of Korea (RS-2025-25456722), and Brain Pool Plus Fellowship Program funded by the Ministry of Science and ICT (RS-2025-25427881). The views expressed are those of the authors and not necessarily those of the NIHR or the Department of Health and Social Care.

## Author contributions

Conceptualization: W.H., D.K., F.M. and N.H. Methodology: W.H., D.K., F.M. and N.H. Investigation: W.H., D.K., W.G., L.C. and S.A. Visualization: W.H., D.K., F.M. and N.H. Supervision: F.M. and N.H. Writing—original draft: W.H., D.K., M.M., F.M. and N.H.Writing—review & editing: W.H., D.K., W.G., M.M., R.H., F.M. and N.H.

## Competing interests

N.H. is a cofounder of KURE.ai and CardiaTec Biosciences. W.H. and M.M. are employees of CardiaTec Biosciences. D.K. is an employee of The Boston Consulting Group, Berlin, Germany. L.C. is an employee of Adaptimmune. R.H. is an employee of Cancer Research Horizons. All other authors declare that they have no competing interests.
