## [Transparent Peer Review file · Communications Chemistry]

Utilising multi-modal data-driven network analysis to identify monotherapy and combinational therapy targets in SOX2-dependent squamous cell lung cancer

Corresponding Author: Professor Namshik Han

Version 0:

Reviewer comments:

Reviewer #1

(Remarks to the Author)

General Comments

The manuscript presents a comprehensive analysis of SOX2-dependent lung squamous cell carcinoma (LUSC) by integrating network-based approaches, drug-target interactions, and prioritization strategies. The study employs the Louvain algorithm on the LUSOX network, followed by clustering on a subset of 168 key proteins, and explores druggability through various computational and experimental techniques. While the manuscript is well-structured and provides valuable insights, certain aspects of figure organization, experimental validation, and methodological clarity require improvement. Below, I outline specific concerns and suggestions for enhancement.

1. Figure Organization and Readability

(1) Louvain Algorithm Iterative Application and Figure Adjustments

The manuscript utilizes the Louvain algorithm on the LUSOX network and then reapplies it to a sub-network of 168 key proteins. While this iterative analysis is crucial for comprehensive understanding, the extensive use of figures and detailed descriptions may dilute the main message. To enhance clarity, I recommend moving process-oriented figures (Figure 3A and Figure 3B) to the Supplementary Information (SI).

(2) Merging Figures 3E and 4A

Both figures present network analysis results, highlighting functional partitioning and drug-target information, respectively. To improve overall readability and coherence, I suggest merging them into a single comprehensive figure.

(3) Relocating Figure 4B (Molecular Similarity Heatmap) to SI

While Figure 4B provides detailed information on drug structural similarity, it contributes to visual complexity in the main text. Moving it to the SI would streamline the main figures and maintain a sharper focus on core research findings.

(4) Enhancing Readability of Drug Names in Figure 4C

The current display format makes drug names difficult to distinguish. Reducing the number of directly labeled drugs on the x-axis and providing a full list in the SI would improve clarity and readability.

2. Experimental and Target-Related Concerns

(1) FOXM1 and KLF5 Experimental Discussion Deficiency

The manuscript does not sufficiently discuss the experimental results for FOXM1 and KLF5, both of which are non-druggable targets. I recommend elaborating on their functional roles in SOX2-dependent LUSC pathogenesis, their interactions with other key proteins, and their relevance in developing combination therapies.

(2) Unclear Justification for TRIM28 as a Selected Target

The manuscript introduces TRIM28 as a selected target without providing clear justification. Previous discussions focus on p63, FOXM1, and KLF5 as meeting selection criteria, but TRIM28 appears abruptly. Additional explanation is needed to clarify why TRIM28 meets the screening criteria and how it differs from or complements other non-druggable targets.

(3) Potential Off-Target Effects and SOX2-Independent Mechanisms

AZD5363, PF-04691502, and JNJ-7706621 are multi-target drugs. It is essential to rule out whether their anticancer effects are unrelated to the SOX2 pathway. Further controls or data are needed to address this concern.

(4) Cell Line Selection and Its Impact on Conclusion Specificity

The inclusion of esophageal cancer cell lines (KYSE series) in the non-SOX2-dependent group, rather than lung squamous cell carcinoma (LUSC) lines, may introduce tissue origin heterogeneity, affecting the determination of SOX2 dependency. Additionally, the absence of a direct comparison between SOX2 high- and low-expressing LUSC cell lines could limit the robustness of the conclusions.

(5) Lack of Mechanistic Validation for SOX2 Downstream Pathways

The study does not provide sufficient data on changes in the expression or phosphorylation levels of SOX2 downstream pathways upon drug treatment. This makes it difficult to demonstrate that drug efficacy is selectively dependent on SOX2.

Additional mechanistic validation is required.

3. Methodology and Data Analysis Issues

(1) Inconsistent iPANDDA Scoring Criteria

The manuscript lacks clarity in the criteria for prioritizing druggable and non-druggable targets using iPANDDA Scoring. Specifically, the RWR score is only applied to non-druggable targets without clear justification. The section "Final Prioritisation Using iPANDDA Scoring" should be reorganized to explain the rationale behind different scoring criteria and their interrelations.

(2) Inconsistencies in OmniPath and STRING Data Sources

OmniPath identifies only 12 direct SOX2 interaction proteins, whereas STRING identifies 156. However, the manuscript does not explain these differences in data sources or filtering criteria. To avoid confusion, I recommend providing a comparison of their selection methodologies and discussing their impact on the study's conclusions. Additionally, if the 12 SOX2-interacting proteins from OmniPath were not used in the final analysis, this information should be removed to prevent ambiguity.

(3) Clarifying Target Selection Criteria in Relation to Figure 4C

Figure 4C highlights key druggable targets and upregulated target proteins, while Figure 4A indicates that multiple downregulated drug targets exist. This raises two key concerns:

Does the emphasis on upregulated targets imply that downregulated ones are less significant for understanding SOX2-dependent LUSC pathogenesis and therapeutic strategies?

The study excludes proteins with downregulated expression in LUSC during target prioritization. A detailed rationale for this exclusion should be provided to help readers better understand the methodology and its implications.

Reviewer #2

(Remarks to the Author)

In this study, the authors present an analytical framework named iPANDDA and report the identification of seven druggable targets associated with SOX2-dependent squamous cell lung cancer. While the authors assert that iPANDDA facilitates the discovery of drug targets, it appears that a significant portion of the target identification process relied heavily on prior biological knowledge—such as information related to SOX2 and data from the STRING database. The iPANDDA method was subsequently employed to refine this list and prioritize biologically plausible targets, many of which have already been supported by experimental evidence. Given this workflow, it remains unclear what unique contribution iPANDDA is expected to make in the broader context of drug discovery. I believe that most researchers in the community would not find the authors' conclusions surprising; even without iPANDDA, these targets would (or already have) been discovered.

I believe that the research community is in need of methodologies capable of identifying novel drug targets from multi-modal datasets, rather than approaches that predominantly retrieve previously reported targets. In this regard, iPANDDA appears to function more as a straightforward, literature-driven approach than as a pipeline designed for the discovery of new biological insights. While I am aware that several studies with similar methodological frameworks exist in this field, I find that such approaches generally fall short of the academic novelty typically expected in Communications Chemistry. Moreover, I believe that researchers of the authors' caliber are expected to contribute more deeply insightful and scientifically innovative work.

Version 1:

Reviewer comments:

Reviewer #1

(Remarks to the Author)

The authors have addressed the necessary points, and I recommend the acceptance of the article.

Reviewers' comments:

Reviewer #1 (Remarks to the Author):

General Comments

The manuscript presents a comprehensive analysis of SOX2-dependent lung squamous cell carcinoma (LUSC) by integrating network-based approaches, drug-target interactions, and prioritization strategies. The study employs the Louvain algorithm on the LUSOX network, followed by clustering on a subset of 168 key proteins, and explores druggability through various computational and experimental techniques. While the manuscript is well-structured and provides valuable insights, certain aspects of figure organization, experimental validation, and methodological clarity require improvement. Below, I outline specific concerns and suggestions for enhancement.

1. Figure Organization and Readability

Q1. Louvain Algorithm Iterative Application and Figure Adjustments

The manuscript utilizes the Louvain algorithm on the LUSOX network and then reapplies it to a sub-network of 168 key proteins. While this iterative analysis is crucial for comprehensive understanding, the extensive use of figures and detailed descriptions may dilute the main message. To enhance clarity, I recommend moving process-oriented figures (Figure 3A and Figure 3B) to the Supplementary Information (SI).

We thank the reviewer for the constructive suggestion regarding the presentation of the Louvain algorithm application. We agree that moving the more process-oriented figures to the Supplementary Information can help streamline the main text and maintain focus on key results.

Accordingly, we have relocated Figures 3A and 3B to Supplementary Figure 2 (now Supplementary Figures 2A and 2B). The main Figure 3 now consists of the former Figures 3C–E, relabeled as Figures 3A–C, respectively. We have updated all corresponding references in the main text to reflect this change.

Q2. Merging Figures 3E and 4A

Both figures present network analysis results, highlighting functional partitioning and drug-target information, respectively. To improve overall readability and coherence, I suggest merging them into a single comprehensive figure.

Thank you for the helpful suggestion regarding the potential merging of Figures 3E and 4A. We agree that improving figure coherence is important. However, we would like to clarify that Figure 4A is not part of the network analysis results shown in Figure 3E. Rather, it presents the outcome of the *in-silico* drug simulation, specifically illustrating the number of predicted drug-target interactions per protein and their expression patterns in TCGA LUSC.

We apologise for the confusion caused by the initial layout. To address this, we have revised the manuscript (lines 274–284 in the section *Network-based druggable target prioritisation and scoring*) to explicitly separate the network analysis from the drug simulation, and to clarify the role of Figure 4A in the overall workflow.

Q3. Relocating Figure 4B (Molecular Similarity Heatmap) to SI

While Figure 4B provides detailed information on drug structural similarity, it contributes to visual complexity in the main text. Moving it to the SI would streamline the main figures and maintain a sharper focus on core research findings.

We thank the reviewer for the helpful suggestion to improve figure clarity and focus. We agree that relocating the molecular similarity heatmap helps streamline the main figures. Accordingly, we have moved Figure 4B to Supplementary Figure 3. As a result, the original Supplementary Figures 3–5 have been renumbered to Supplementary Figures 4–6, and all relevant references in the text have been updated accordingly.

Q4. Enhancing Readability of Drug Names in Figure 4C

The current display format makes drug names difficult to distinguish. Reducing the number of directly labeled drugs on the x-axis and providing a full list in the SI would improve clarity and readability.

We thank the reviewer for this helpful suggestion to improve the readability of Figure 4C. In response, we have reduced the number of directly labelled drugs in the x-axis of Figure 5C to minimise visual clutter and enhance clarity. The full list of drug names has been provided in Supplementary Figure 7C for reference.

2. Experimental and Target-Related Concerns

Q5. FOXM1 and KLF5 Experimental Discussion Deficiency

The manuscript does not sufficiently discuss the experimental results for FOXM1 and KLF5, both of which are non-druggable targets. I recommend elaborating on their functional roles in SOX2-dependent LUSC pathogenesis, their interactions with other key proteins, and their relevance in developing combination therapies.

We thank the reviewer for pointing this out. We have amended the text as suggested (line 409).

Q6. Unclear Justification for TRIM28 as a Selected Target

The manuscript introduces TRIM28 as a selected target without providing clear justification. Previous discussions focus on p63, FOXM1, and KLF5 as meeting selection criteria, but TRIM28 appears abruptly. Additional explanation is needed to clarify why TRIM28 meets the screening criteria and how it differs from or complements other non-druggable targets.

We thank the reviewer for highlighting the need for clarification regarding the selection of TRIM28 as a therapeutic target. We recognize that the initial manuscript text may have inadvertently created confusion regarding the classification of TRIM28. To clarify explicitly: TRIM28 was not categorized as a non-druggable target. Instead, TRIM28 was selected as a transcription factor (TF) candidate specifically for combinatorial therapeutic strategies.

Below we provide a detailed rationale and clarify the relationship between TRIM28 and the other target proteins mentioned:

1. Clarification on Target Categories

- TRIM28 was selected as a community-defining transcription factor candidate for combination therapy based on its regulatory role within the WNT-SOX2 network.
- In contrast, other candidate proteins (p63, FOXM1, and KLF5) were independently identified and prioritized as non-druggable monotherapy targets based on global filtering criteria (gene expression in LUSC samples, dependency scores from DepMap)

data, and direct interactions with SOX2). These proteins were not combined with TRIM28.

2. Rationale for TRIM28 Selection and Combination Therapy

Our selection of TRIM28 was driven by a systematic network-based approach:

- Initially, we identified four functional protein communities (cell cycle, cytokine signaling, PI3K-AKT pathway, and WNT pathway) using community detection methods. Within the WNT pathway community containing SOX2, we conducted transcription factor enrichment analysis using the ENCODE TF binding site dataset via Enrichr (see Methods: Identification of combination therapy).
- Through this analysis, TRIM28 emerged as the most significantly enriched TF, directly regulating eight proteins interacting with SOX2 (Figure 5A). Given its central regulatory role, we selected TRIM28 as a strategic TF candidate for combination therapy.
- Following established network medicine principles suggesting enhanced efficacy from combinations targeting distinct communities [1], we evaluated TRIM28 knockout in combination with inhibition of key druggable proteins from other communities—CDK1 (cell cycle), AKT (cytokine signaling), and FGFR (PI3K-AKT pathway). This intentional cross-community targeting approach aims to exploit complementary disease pathways to enhance therapeutic outcomes.

Combination therapies involving transcription factor inhibition are well-established in cancer therapeutics, providing further validation for our approach [2-4].

In response to the reviewer's insightful comments, we have explicitly revised the manuscript text in the section "Prioritisation and Validation of Non-Druggable Targets and Combination Therapies" to clarify the categorization and rationale for selecting TRIM28 as a combination therapy candidate distinct from other non-druggable targets.

We hope this revised explanation addresses the reviewer's concerns clearly and comprehensively.

References

1. Cheng, F., Kovács, I.A. & Barabási, AL. Network-based prediction of drug combinations. *Nat Commun* **10**, 1197 (2019). <https://doi.org/10.1038/s41467-019-09186-x>
2. Austin G. Holmes *et al* .,A MYC inhibitor selectively alters the MYC and MAX cistromes and modulates the epigenomic landscape to regulate target gene expression. *Sci. Adv.* **8**, eabh3635(2022). DOI:[10.1126/sciadv.abh3635](https://doi.org/10.1126/sciadv.abh3635)
3. Yu, H., Lin, L., Zhang, Z. *et al*. Targeting NF-κB pathway for the therapy of diseases: mechanism and clinical study. *Sig Transduct Target Ther* **5**, 209 (2020). <https://doi.org/10.1038/s41392-020-00312-6>
4. Yu, H., Pardoll, D. & Jove, R. *et al*. STATs in cancer inflammation and immunity: a leading role for STAT3. *Nat Rev Cancer* **9**, 798–809 (2009). <https://doi.org/10.1038/nrc2734>

Q7. Potential Off-Target Effects and SOX2-Independent Mechanisms

AZD5363, PF-04691502, and JNJ-7706621 are multi-target drugs. It is essential to rule out whether their anticancer effects are unrelated to the SOX2 pathway. Further controls or data are needed to address this concern.

We thank the reviewer for raising this important issue. SOX2 is a pleiotropic oncogene which dramatically impacts the cellular transcriptional and epigenetic landscape. As a result, we do

not consider there is a specific SOX2 pathway that can be interrogated. Moreover, the impact of a specific kinase inhibitor on a putative SOX2 pathway would be very difficult to predict. We see iPANDDA as network-based approach to infer potential therapeutic vulnerabilities associated with SOX2 dependency; in this scenario the key experiments are those presented, that is the impact on cell population expansion in SOX2 vs non-SOX2 dependent cell lines. As the reviewer rightly observes kinase inhibitors can affect multiple targets, in particular PF-04691502. We therefore now include dose-response data from other mTOR inhibitors assayed against the same cell lines with very similar results, emphasizing the differential impact on SOX2-dependent compared to non-SOX2-dependent cell lines – data presented in Supplementary Figure 5E. A reference to these data is included (line 377).

Q8. Cell Line Selection and Its Impact on Conclusion Specificity

The inclusion of esophageal cancer cell lines (KYSE series) in the non-SOX2-dependent group, rather than lung squamous cell carcinoma (LUSC) lines, may introduce tissue origin heterogeneity, affecting the determination of SOX2 dependency. Additionally, the absence of a direct comparison between SOX2 high- and low-expressing LUSC cell lines could limit the robustness of the conclusions

The reviewer raises an important issue in the field: the paucity of useful cell lines for interrogating therapeutic targets for LUSC and highlighted in Supplementary Figure 5. This is compounded by the particular challenges associated with establishing patient-derived xenografts and mouse models of squamous lung cancer. The canonical histological markers of squamous lung cancer are KRT5 and p63 (p40) positivity. Key LUSC cell lines with SOX2-dependency do not express the canonical trio of markers of (SOX2, p63 (p40), KRT5). We therefore were keen to include a SOX2-dependent squamous cell line that expresses each of these markers and chose KYSE-140. It remains an unexplained fact why it has proven so challenging to generate LUSC cell lines.

Although it is of course impossible to exclude an issue associated with tissue of origin heterogeneity, we are reassured by the robust association between SOX2 and squamous cancers, and the growing body of evidence that biomarker-driven cancer therapeutic decisions are made on a tissue agnostic basis. We have addressed this and included an appropriate reference in the Discussion (line 477).

Q9. Lack of Mechanistic Validation for SOX2 Downstream Pathways

The study does not provide sufficient data on changes in the expression or phosphorylation levels of SOX2 downstream pathways upon drug treatment. This makes it difficult to demonstrate that drug efficacy is selectively dependent on SOX2. Additional mechanistic validation is required.

Many thanks for this point and the opportunity for clarification. However, we do not feel that it is possible to address this in the way suggested. The SOX2-dependency is an indirect target of the kinase inhibitors used as SOX2 itself is not targeted. As we note in the answer to point Q7, there is no specific SOX2 pathway, rather this is both a promiscuous and pleiotropic transcription factor that has a huge interactome (Figure 3). It is not possible to predict the impact and timing of impact of specific kinase inhibition in heterogenous cell lines. Rather, iPANDDA suggests that the SOX2-dependency “network” should be vulnerable to specific kinase inhibition and the results in Figure 4D and Supplementary Figure X are consistent with that. We show a differential vulnerability to specific kinase inhibition associated with SOX2 status.

We can demonstrate target engagement and a differential response to kinase inhibitors between SOX2 dependent and independent cell lines (Supplementary Figure 5E) but, of

course, cannot relate this to exclusively SOX2 downstream pathways. We have amended the discussion to emphasise the way we interpret the results of iPANDDA.

Rebuttal Figure 1: Downstream pathway response to AKT inhibition

The indicated cell lines were cultured for 72h with 3 μ M AZD5363. An anticipated impact of AZD5363 treatment is hyperphosphorylation of AKT. This is demonstrated and is associated a reduction in pS6 (that generally sits downstream of AKT) in 2 of 3 SOX2-dependent cell line and an increase in pS6 in 2 of 3 non-SOX2-dependent cell lines. It is likely in H520 that the AKT signalling cascade is via alternative downstream mediators.

3. Methodology and Data Analysis Issues

Q10. Inconsistent iPANDDA Scoring Criteria

The manuscript lacks clarity in the criteria for prioritizing druggable and non-druggable targets using iPANDDA Scoring. Specifically, the RWR score is only applied to non-druggable targets without clear justification. The section "Final Prioritisation Using iPANDDA Scoring" should be reorganized to explain the rationale behind different scoring criteria and their interrelations.

Thank you for your comment regarding the use of iPANDDA Scoring criteria for prioritizing druggable and non-druggable targets.

Network proximity in our drug simulations is based on compound–target interaction data. For non-druggable targets, such interaction data were not available, making it infeasible to calculate a network proximity score. Therefore, only the random walk restart (RWR) score from the SOX2 node was used for their prioritization. This clarification has now been added to both the Results section ("Prioritisation and Validation of Non-Druggable Targets and Combination Therapies") and the Methods section ("Prioritisation of key proteins for therapy").

Q11. Inconsistencies in OmniPath and STRING Data Sources

OmniPath identifies only 12 direct SOX2 interaction proteins, whereas STRING identifies 156. However, the manuscript does not explain these differences in data sources or filtering criteria. To avoid confusion, I recommend providing a comparison of their selection methodologies and discussing their impact on the study's conclusions. Additionally, if the 12 SOX2-interacting proteins from OmniPath were not used in the final analysis, this information should be removed to prevent ambiguity.

We thank the reviewer for this important observation. We acknowledge that the differences between the number of SOX2-interacting proteins identified by STRING (n = 156) and OmniPath (n = 12) may appear inconsistent and could lead to confusion if not properly contextualized.

To clarify:

- All downstream network analyses, including protein–protein interaction (PPI) network construction, community detection, and target prioritisation, were conducted using the STRING-derived interaction network. STRING provides an integrative dataset that includes experimentally validated and computationally predicted interactions, enabling a broad and systems-level view of SOX2-related connectivity in LUSC.
- The 12 direct SOX2-interacting proteins from OmniPath (accessed via OpenTargets) represent a stringently curated subset of high-confidence physical interactions. These proteins are a subset of the STRING-derived 156 interactions and were not used separately or additionally in any analytical step.
- We have clarified in the revised manuscript that the 12 OmniPath interactions were mentioned to indicate concordance with STRING results and that they are fully included within the STRING dataset used for analysis. We also revised the text to avoid any implication that these 12 proteins were used as an independent dataset or analytical input.
- As suggested, we now explicitly explain the differences in curation approaches between the two databases. STRING integrates a wide range of evidence types (experimental, database, co-expression, predictive models), whereas OmniPath relies on manually curated, literature-supported direct physical interactions. This distinction helps explain the disparity in interaction counts and supports the interpretation that the STRING-derived SOX2 interactome includes both broad and high-confidence interactions.

We have incorporated this clarification in the revised manuscript in the section “Construction of LUSC SOX2-dependent (LUSOX) network” (line 184). This revision ensures transparency about data source usage and eliminates potential confusion regarding the role of OmniPath-derived interactions.

We believe these clarifications resolve the ambiguity and reinforce the rigor of our network construction approach.

Q12. Clarifying Target Selection Criteria in Relation to Figure 4C

Figure 4C highlights key druggable targets and upregulated target proteins, while Figure 4A indicates that multiple downregulated drug targets exist. This raises two key concerns: Does the emphasis on upregulated targets imply that downregulated ones are less significant for understanding SOX2-dependent LUSC pathogenesis and therapeutic strategies?

The study excludes proteins with downregulated expression in LUSC during target prioritization. A detailed rationale for this exclusion should be provided to help readers better understand the methodology and its implications.

Thank you for your comment regarding our approach to target prioritization.

As described in the revised Methods section, we excluded proteins with downregulated expression in LUSC samples from the TCGA dataset. Figure 4C presents the expression profiles of drug targets in LUSC for compounds included in our *in-silico* simulations. Notably, analysis of the predicted compounds' mechanisms of action showed that the vast majority (>90%) function as inhibitors (Supplementary Figure 3B). Since the major mechanism of action of the predicted compounds is inhibition, we excluded targets with decreased expression in LUSC from further consideration.

This rationale is now explicitly stated in the Methods section ("Prioritisation of key proteins for therapy"), with reference to the relevant supplementary figure.

We thank the reviewer for prompting us to clarify this methodological decision.

Reviewer #2 (Remarks to the Author):

In this study, the authors present an analytical framework named iPANDDA and report the identification of seven druggable targets associated with SOX2-dependent squamous cell lung cancer. While the authors assert that iPANDDA facilitates the discovery of drug targets, it appears that a significant portion of the target identification process relied heavily on prior biological knowledge—such as information related to SOX2 and data from the STRING database. The iPANDDA method was subsequently employed to refine this list and prioritize biologically plausible targets, many of which have already been supported by experimental evidence. Given this workflow, it remains unclear what unique contribution iPANDDA is expected to make in the broader context of drug discovery. I believe that most researchers in the community would not find the authors' conclusions surprising; even without iPANDDA, these targets would (or already have) been discovered.

I believe that the research community is in need of methodologies capable of identifying novel drug targets from multi-modal datasets, rather than approaches that predominantly retrieve previously reported targets. In this regard, iPANDDA appears to function more as a straightforward, literature-driven approach than as a pipeline designed for the discovery of new biological insights. While I am aware that several studies with similar methodological frameworks exist in this field, I find that such approaches generally fall short of the academic novelty typically expected in Communications Chemistry. Moreover, I believe that researchers of the authors' caliber are expected to contribute more deeply insightful and scientifically innovative work.

We thank the reviewer for their time and insightful comments on our manuscript describing iPANDDA. We appreciate the opportunity to address the concerns raised, particularly regarding the novelty and unique contribution of our proposed methodology.

The reviewer suggests that iPANDDA may rely heavily on prior biological knowledge and function more as a literature-driven approach to refine existing findings, rather than a pipeline for discovering novel biological insights. We respectfully disagree with this assessment and wish to clarify the unique aspects of iPANDDA.

iPANDDA is not simply an aggregation of existing databases or a mere refinement tool for previously reported targets. Instead, it is designed as an independent and distinct

methodology specifically for the identification of disease-specific drug targets. While iPANDDA, like many computational biology tools, leverages existing biological knowledge (such as information from STRING or data related to SOX2) as a foundational layer, it applies a novel, multi-modal integration strategy to identify context-specific targets that are not retrievable by any single input source alone. Its core strength lies in its unique analytical framework that integrates and processes this information in a novel way to pinpoint targets pertinent to a specific pathological context.

To validate the originality and efficacy of iPANDDA, we conducted comprehensive comparative analyses. Crucially, we benchmarked iPANDDA against both internal datasets and established external methodologies (Rebuttal Figure 2). These evaluations demonstrated significant differences between the target candidates identified by iPANDDA and those from other approaches. For instance, to highlight these differences, we selected our top 10 high-confidence candidates and measured the recall of this set within the top-ranked lists of other major databases. The analysis revealed a notably low recall from established methods: OpenTargets recalled only 5 and 4 of the 10 genes (a recall of 0.5 and 0.4), CTD achieved a recall of 0.5, and DisGeNET only 0.1. This poor recall by other methods underscores that iPANDDA is not merely recapitulating known information; rather, it is identifying a unique set of targets that might be missed or de-prioritized by them.

Furthermore, the utility of iPANDDA extends beyond this top 10 benchmark, demonstrating its capacity to generate deeper biological insights from its comprehensive results. By establishing a longer, prioritized list of candidates within a defined interactome, our framework uncovers other valuable therapeutic opportunities. A key example is the identification of TRIM28. While not part of the top 10 validation set, TRIM28 was a high-ranking candidate from our broader analysis that represents a plausible but underexplored target for this disease. The discovery of such targets is crucial as it provides a robust foundation for designing rational combination therapies. This capability to identify not only the most prominent targets but also a wider set of plausible candidates for synergistic strategies is a key innovative aspect of iPANDDA.

In summary, iPANDDA is presented as a scientifically innovative methodological development. Its novelty is supported by distinct lines of empirical evidence: **1)** quantitative benchmarking showing its top-ranked outputs are unique, and **2)** its ability to identify other high-potential candidates like TRIM28 from its broader results. Contrary to the reviewer's suggestion that these targets "would (or already have) been discovered," our analysis provides clear evidence that iPANDDA highlights a distinct set of underexplored targets. This capacity for generating genuinely novel findings, which in turn opens avenues for advanced therapeutic strategies like combination approaches, directly addresses the need for methodologies that provide deeper biological insights.

Thank you for your reconsideration.

Rebuttal Figure 2: Comparative benchmark analysis of iPANDDA's high-confidence targets

The bar plot evaluates the recall of the top 10 high-confidence candidate genes identified by iPANDDA. The analysis measured how many of these 10 genes (red bar) were also present within the top-ranked results of internal datasets, including OpenTargets (v19.9), DepMap, TCGA-LUSC, and SOX2 RNA-seq—as well as external resources such as OpenTargets (v23.3), CTD, and DisGeNET. The results demonstrate a low recall rate across all other platforms, with even the best-performing methods like OpenTargets and CTD failing to identify half of the candidate list. Importantly, the overlap between iPANDDA and each individual dataset was minimal (e.g., <50% shared with OpenTargets), indicating that iPANDDA does not simply recapitulate known targets. This result underscores the methodological distinctiveness of iPANDDA and its capacity to identify disease-relevant drug targets not captured by conventional literature- or data-driven approaches. These findings support the view that iPANDDA offers added value in identifying biologically meaningful and previously underexplored therapeutic candidates.